# Identification of mobile development issues using semantic topic modeling of Stack Overflow posts



Fatih Gurcan

Department of Management Information Systems, Faculty of Economics and Administrative Sciences, Karadeniz Technical University, Trabzon, Turkey

## ABSTRACT

**Background:** Increasing demands for mobile apps and services have recently led to an intensification of mobile development activities. With the proliferation of mobile development, there has been a major transformation in the architectures, paradigms, knowledge domains and skills of traditional software systems towards mobile development. Therefore, mobile developers experience a wide spectrum of issues specific to development processes of mobile apps and services.

**Methods:** In this article, we conducted a semantic content analysis based on topic modeling using mobile-related questions on Stack Overflow, a popular Q&A site for developers. With the aim of providing an understanding of the issues and challenges faced by mobile developers, we used a semi-automated methodology based on latent Dirichlet allocation (LDA), a probabilistic and generative approach for topic modeling.

**Results:** Our findings revealed that mobile developers' questions focused on 36 topics in six main categories, including "Development", "UI settings", "Tools", "Data Management", "Multimedia", and "Mobile APIs". Besides, we investigated the temporal trends of the discovered issues and their relationships with mobile technologies. Our findings also revealed which issues are the most popular and which issues are the most difficult for mobile development. The methodology and findings of this study have valuable implications for mobile development stakeholders including tool builders, developers, researchers, and educators.

## INTRODUCTION

In today's digital world, with the spread of mobile devices and technologies, the demands for mobile-oriented services and apps in all industrial and social fields are increasing exponentially day by day. Every day, a large number of mobile apps are presented for users in the Android Play store and Apple App store (*Jabangwe, Edison & Duc, 2018*). This strong increase in demand for mobile services and apps has led to the intensification of mobile software development activities and the emergence of mobile software engineering as a contemporary discipline (*Ahmad et al., 2018b*). The advent of mobile software engineering has also led to a major transformation in traditional software engineering architectures, methodologies, knowledge domains, and skills (*Dhillon & Mahmoud, 2015*; *Nagappan & Shihab, 2016*; *Ahmad et al., 2018b*). For this reason, an increasing number of

Corresponding author
Fatih Gurcan, fgurcan@ktu.edu.tr

software developers, practitioners and researchers discuss the issues encountered in mobile app development processes and seek solutions to these problems (*Rosen & Shihab, 2016*; *Ahmad et al., 2018b*). As a natural consequence of this situation, mobile development issues have become popular topics that have been increasingly discussed recently on question and answer (Q&A) platforms like Stack Overflow (*Stack Overflow, 2023*).

Stack Overflow is the most common Q&A platform that manages millions of posts from developers with different backgrounds in mobile development or different specialties to share their technical issues and find solutions to them (*Ahmad et al., 2018a*; *Beyer et al., 2020*). Any sub-context of these posts on Stack Overflow provides a comprehensive data repository that includes important information, such as developers' problems, solutions, and perspectives specific to that context (*Beyer et al., 2020*; *Hu et al., 2020*). Analysis of the different sub-contexts of this data repository can provide remarkable insights into understanding and solving problems focused on by developers in different specialties (*Rosen & Shihab, 2016*; *Ahmad et al., 2018a*). From this point of view, many researchers have conducted experimental studies to investigate specific sub-contexts in Stack Overflow, such as testing, security, mobile development, requirements, concurrency development, chatbot development, and machine learning (*Vasilescu, 2014*; *Ahmad et al., 2018a*; *Hu et al., 2020*). On the other hand, the number of studies on mobile development using Stack Overflow data is relatively limited. The previous studies on mobile development included specific sub-contexts such as Android testing, Android development, iOS development, and main themes of mobile development (*Rosen & Shihab, 2016*; *Ahmad et al., 2018a*; *Jabangwe, Edison & Duc, 2018*). The aforementioned research, which were conducted prior to 2016, do not provide current perspectives on the challenges encountered by mobile developers due to the significant surge in mobile usage in recent years, particularly in the aftermath of the COVID-19 pandemic (*Gurcan, Dalveren & Derawi, 2022*; *Stack Overflow, 2022*). With the integration of cutting-edge technologies such as immersion, artificial intelligence, IoT, blockchain, wearable devices, and cloud services into mobile apps, mobile development processes and architectures have been experiencing a dizzying evolution recently (*Al-Razgan et al., 2021*; *Gurcan et al., 2022b*). Considering the rapidly exploding demand for mobile apps and services over the last few years, we anticipate that analysis of the developer discussions and solution proposals using Stack Overflow posts will offer significant and up-to-date implications for researchers and practitioners.

Against this background, this study aimed to reveal in depth the problems, interests, and trends of mobile developers in the last 5 years between 2017–2021, thus closing this gap in the literature. From this point of view, Stack Overflow, the most popular online Q&A platform for developers, was chosen as the data source for this study and Latent Dirichlet Allocation (LDA) based topic modeling analysis was applied on the mobile-related posts obtained from it. This proposed systematic methodology included an ordered sequence of semantic topic modeling processes based on LDA (a probabilistic topic modeling algorithm used to reveal latent topics). In this way, mobile developers' topics about their problems and discussions, their underlying dependencies, and trends over time were

identified. More specifically, the methodology of this study was designed to seek answers to the following research questions (RQ):

**RQ1.** What issues are asked about mobile development?
**RQ2.** What are the most difficult and most popular issues for mobile development?
**RQ3.** How have the trends of mobile issues changed over time?
**RQ4.** What are the most asked mobile technologies?
**RQ5.** How have the trends of mobile technologies changed over time?

## BACKGROUND AND RELATED WORK

We base the background of our study on three fundamental pillars and discuss the related work here under the headings including mobile development, Stack Overflow, and Latent Dirichlet Allocation.

### Mobile development

Mobile developers design, develop, and execute applications for smartphones and other mobile devices. They usually develop mobile applications on a specific type of operating system, such as Android, iOS, or on cross-platforms (*Nagappan & Shihab, 2016*). Therefore, mobile development contains different paradigms and methodologies from those in traditional software engineering (*Elsayed et al., 2019*; *Gurcan et al., 2022b*). In the process of developing mobile applications that have become a part of our lives, mobile developers face a wide spectrum of experiences, issues, and challenges unlike common developer issues (*Ahmad et al., 2018b*). From this perspective, the processes, experiences, and practices of mobile development contain domain-specific issues and challenges faced by the developers (*Nagappan & Shihab, 2016*).

Given the issues and challenges of mobile development, we have observed that the studies so far has focused on issues and challenges in specific contexts of mobile development such as platform-specific issues including native, cross-platform, or hybrid development (*El-Kassas et al., 2017*), mobile cloud computing (*Malik et al., 2021*), development process and life-cycle (*Jabangwe, Edison & Duc, 2018*), testing (*Zein, Salleh & Grundy, 2016*), usability and UI design (*Taba et al., 2017*), app security and privacy (*Gurcan et al., 2022a*), tools and frameworks (*El-Kassas et al., 2017*).

In a more specific outlook, with the aim of identifying mobile development issues, a number of empirical studies were conducted, partly similar to our current study. In an empirical study, *Rosen & Shihab (2016)* analyzed the Stack Overflow data dump using a topic modeling approach to examine what mobile developers are asking about. They revealed 40 topics and 32 different category mapping mobile development issues. In a study with a similar perspective, *Linares-Vásquez, Dit & Poshyvanyk (2013)* implemented topic modeling to Stack Overflow data dump to extract the trending topics from mobile development questions. *Beyer & Pinzger (2014)* performed a manual categorization of Android app development issues on Stack Overflow considering problem types. *Villanes et al. (2017)* conducted a study using Stack Overflow data in order to analyze and cluster the main topics on Android testing. *Ahmad et al. (2019)* identified the topics and trends of non-functional requirements for development of iOS applications using Stack Overflow

data. Also, *Fontão et al. (2018)* explored main topics and indicators in the mobile software ecosystem by analyzing technical questions about mobile platforms on Stack Overflow.

Apart from the aforementioned studies using stack overflow data, in a qualitative study, mobile challenges were identified through a systematic literature review and then validated by interviewing practitioners (*Ahmad et al., 2018b*). Besides, *Pandey, Litoriya & Pandey (2018)* identified 14 mobile issues, and using an interpretive structure modeling (ISM) approach, categorized them into four groups: dependent, driving, linkage and autonomous.

Furthermore, researchers have so far conducted many empirical studies using Stack Overflow data in order to shed light on many aspects of software development issues and challenges (*Ahmad et al., 2018a*). In particular, some remarkable studies which use Stack Overflow data have been carried out in order to discover main issues and challenges in specific sub-domains of software development such as testing (*Kochhar, 2016*), security (*Yang et al., 2016*), mobile development (*Linares-Vásquez, Dit & Poshyvanyk, 2013*; *Rosen & Shihab, 2016*), programming languages (*Chakraborty et al., 2021*), requirements (*Zou et al., 2017*), concurrency development (*Ahmed & Bagherzadeh, 2018*), IOT development (*Uddin et al., 2021*), and machine learning (*Alshangiti et al., 2019*). Beyond aforementioned studies, a comprehensive collection of research which uses Stack Overflow data from 2009 to date is provided in more detail by Vasilescu on Stack Exchange Meta (*Vasilescu, 2014*). Also, *Ahmad et al. (2018a)* conducted a comprehensive literature review and categorized 166 research articles (from 2008 to June 2016) using Stack Overflow data. In summary, our current study complements the aforementioned work since our methodology focuses on in-depth analysis of the mobile-related posts shared on Stack Overflow in order to identify mobile development issues and their trends.

## Latent Dirichlet allocation

Topic modeling encompasses a set of methods, procedures, and tools that enable the discovery of hidden semantic structures, called topics, in large collections of textual information (*Blei, 2012*). In text mining and natural language processing implementations, topic modeling approaches widely used for semantic context analysis of the document collections. In topic modeling algorithms, latent Dirichlet allocation (LDA) is a generative probabilistic approach widely used for topic modeling of document collections (*Blei, Ng & Jordan, 2003*). The intuitive idea behind LDA is based on the assumption that each document is characterized by more than one topic, and each topic is characterized by the distribution of words in an empirical *corpus*. LDA treats the words and documents observed in a *corpus* as being created by an underlying topic structure.

It is a difficult process to obtain the posterior distribution by computation in extracting the hidden topic structure of the documents. Therefore, various techniques have been developed for approximate inference, including Gibbs sampling (*Griffiths & Steyvers, 2004*) and variational Bayes approximation (*Blei, Ng & Jordan, 2003*). Each of the mentioned inference techniques possesses distinct advantages and disadvantages that are traded off in terms of their speed, complexity, accuracy, and simplicity (*Vayansky & Kumar, 2020*; *Gurcan, 2023*).

Since the LDA model is based on unsupervised machine learning, it enables the discovery of semantic topics in a short time without the need for any training process (*Blei & Lafferty, 2007*; *Gurcan & Cagiltay, 2022*). Beyond textual data, the LDA model can be effectively applied to different types of data, such as genetic data, software codes, images, videos, forums, blogs, and social networks (*Blei, 2012*; *Silva, Galster & Gilson, 2021*). Because of these supportive features, the LDA model is considered by many authorities as a robust and efficient approach for semantic content analysis that automates the detection of latent topics in the textual contents of a huge *corpus* (*Blei, 2012*; *Gurcan et al., 2022b*).

From its emergence to the present, the LDA model is also often used in software engineering research to analyze structured or unstructured data in software repositories, such as natural language texts, web archives, log files, source codes, mailing list archives, bug reports, Git repositories, Q&A posts, and requirements documents (*Silva, Galster & Gilson, 2021*; *Gurcan et al., 2022a*; *Gurcan, 2023*). Considering the studies specific to mobile development, a number of studies used to the LDA model to investigate mobile development issues asked on Stack Overflow (*Linares-Vásquez, Dit & Poshyvanyk, 2013*; *Rosen & Shihab, 2016*; *Villanes et al., 2017*; *Fontão et al., 2018*; *Ahmad et al., 2019*); to analyze users' feedback, reviews and ratings for mobile apps (*McIlroy et al., 2016*; *Hu et al., 2019*; *Noei et al., 2019*); to extract features from mobile app descriptions and recommend new features for the similar apps (*Jiang et al., 2019*); to detect permission reauthorization vulnerabilities in Android apps (*Demissie, Ceccato & Shar, 2020*); and to reveal the usage of common interface elements in Android apps (*Taba et al., 2017*); and to distinguish malicious Android apps (*Yang et al., 2017*).

As seen from the aforementioned studies, the LDA-based topic modeling approach is widely used in software engineering research. Considering this background, *Silva, Galster & Gilson (2021)* conducted a comprehensive literature review study which revealed the usage of topic modeling in software engineering research. From a similar point of view, *Chen, Thomas & Hassan (2016)* performed a survey on the use of topic models when mining software repositories. Apart from these studies, it is possible to talk about the existence of a large number of the studies based on LDA. As a result, the effectiveness and suitability of this topic model approach for software engineering research further increases our motivation to use the LDA model for investigating mobile development issues.

## METHOD

### Data collection and extraction

With the intention of achieving an objective methodology, we used the Stack Overflow data dump, which is publicly available as XML files (*Internet Archive, 2023*). In the first step, we downloaded the SO data dump in XML format (posts.xml, last updated March 21, 2022) and parsed it into a PostgreSQL database. This parsed data file contained a total of 55,027,254 (22,100,401 questions and 32,926,853 answers) posts from July 2008 to March 2022. Each post in the data dump contains various metadata elements such as title, body, tags, and so on. The datasets generated during and analyzed during the current study are publicly available in the Internet Archive repository (*Internet Archive, 2023*), as a data dump including XML files.

## Identification of mobile-related posts

Posts on Stack Overflow cover a wide range of expertise, experience, and knowledge-domains for developers. Considering that the posts recorded in the database may be related to any specific subject, in this study, we are only interested in mobile related posts and we aim to extract posts within this scope. From this perspective, we have endeavored to put forward an effective and methodological approach to detect only mobile development related posts in a systematic way. From this perspective, we aimed to present an effective and objective approach to identify only mobile-related posts. To achieve this, we created the first draft list containing the keywords related to mobile development and presented them in Table A1.

At this stage, we identified the primary mobile keywords, taking into account previous studies (*Linares-Vásquez, Dit & Poshyvanyk, 2013*; *Rosen & Shihab, 2016*) and Stack Overflow's annual developer surveys (*Stack Overflow, 2022*). Although the list in Table A1 does not include all mobile-related keywords, as a first draft, these keywords consist of fundamental components of mobile development such as operating systems, hardware, development platforms and SDKs. Namely, beyond the mobile-related posts obtained using these initial keywords, we envisaged that mobile-related tags include a wider range and there may be many mobile tags that are not included in Table A1. Therefore, we performed a number of sequential procedures to identify additional mobile-related tags in a systematic way. Initially, we identified all Stack Overflow posts that contain any of the first keyword listed in Table A1. Then, we extracted the tags for each of these posts. The tags represent keywords that users associate with their questions. Thus, we extracted all the tags for these mobile posts and obtained a larger set of tags. This approach we used allowed us to discover new tags and thus get a richer set of mobile posts. On the other hand, the drawback of this approach was that it could also include a large number of posts not related to mobile in the dataset. For example, let's consider a post with tags "android", "testing", "unit-testing", and specify three tags from that post. Let us say we then include all posts that contain any of these three tags. In such a case, some posts may be related to the testing process of any desktop or web application, even though they have the "testing" tag.

In another example, although Java is a common language for Android apps, posts with the tag "java" may not always be mobile related. Because Java is used in many other types of applications other than mobile. Because, Java is also used for many platforms other than mobile. Therefore, many of the posts with the "java" tag may be related to many other development issues besides mobile. In such cases, adding all posts having a "java" tag would cause unrelated posts with mobile to be included in the dataset. Accordingly, this process would lead to a significant noise in the dataset.

In order to overcome this problem, we employed a set of procedures based on quantitative approaches. In the first step, we extract all the tags of the posts containing the keywords in Table A1 and define these tags as candidate tags ($C_t$). In this stage, we aimed to identify mobile-related ones among these candidate tags and to obtain more tags. In order to identify only mobile-related ones among the candidate tags, and to calculate how relevant they are to mobile, we defined three variables $Var^A$, $Var^B$, and $Var^C$ for each ($C_t$)

 

candidate tag. Specifically, $Var^A$ indicates the number of posts that contain both the candidate tag ($C_t$) and at least one of the mobile keywords in Table A1 within their tags. $Var^B$ indicates the number of posts that contain the candidate tag ($C_t$) in their tags among all posts. Considering $Var^A$ and $Var^B$, we defined a tag relevance score ($TRS_t$) for each candidate tag ($C_t$) as follows:

$$TRS_t = \frac{Var^A}{Var^B}$$

$TRS_t$ indicates how relevant the candidate tag ($C_t$) is to mobile. The value of $TRS_t$ ranges from 0 to 1. The greater the value of $TRS_t$, the more relevant the candidate tag ($C_t$) is to mobile. The case where the value of $TRS_t$ is equal to 1 indicates that the candidate tag ($C_t$) tag appears only with the mobile tags in Table A1. With this in mind, we performed a number of experiments with different $TRS_t$ values. We manually evaluated the tags listed as the output of each experiment and concluded that using the $TRS_t$ value of 53% produced optimal results without being too restrictive. After excluding irrelevant tags using $TRS_t$, we detected very low-importance tags that were related to a very rare issue that only appeared in one or two posts. The value of $TRS_t$ is 1 for a candidate tag ($C_t$) that only appears on a post (*e.g.*, "android-iconics", "android-content-capture" or "android-device-controls"). In this case, the inclusion of such low frequency tags will lead to an increase in the amount of unimportant data in the dataset. In order to solve this problem, we set another threshold value called tag significance score ($TSS_t$) for each candidate tag ($C_t$) and calculated as follows:

$$TSS_t = \frac{Var^A}{Var^C}$$

In this formula, $Var^C$ is the number of mobile posts containing the most popular mobile tag. In the *corpus* of our study, the tag "android" was the most common, contained within 465,178 posts (from 2017 to 2021). We tried different $TSS_t$ values and evaluated the tags listed as the outputs of each experiment. After that, we concluded that the inclusion of candidate tags ($C_t$) with $TSS_t$ values of 0.5% and above gives optimal results. Finally, the list of 66 identified tags, and their $TRS_t$ and $TSS_t$ values are given in Table A2. The tags in Table A2 are sorted by $TSS_t$ in descending order. Similar approaches to the one we used to identify mobile-related tags have also been used in previous studies for investigation of different sub-contexts of Stack Overflow (*Rosen & Shihab, 2016*; *Yang et al., 2016*; *Uddin et al., 2021*).

## Creation of empirical corpus of mobile posts

In this study, we aimed to reveal the landscape of themes and trends in mobile development in more detail, especially in recent years, so we included the posts covering the last 5 years from January 1, 2017 to January 1, 2022 in our experimental dataset. To this end, we tried to extract mobile-related posts shared in the last 5 years using the final set of mobile tags given in Table A2. Initially, we identified all question posts containing the tags in Table A2 within the tags assigned to each question post. Next, we extracted the answers

and descriptive indicators (title, body, creation date, favorite count, comment count, view count, score, answer count, *etc.*) of these questions. These extracted question and answer posts constitute our final empirical dataset that we will use in our experiments. In total, our empirical *corpus* contains 2,242,504 posts including 1,036,682 questions and 1,205,822 answers. The monthly distribution of the number of Stack Overflow posts over the last 5 years is given in Fig. A1. According to Fig. A1, it is observed that the number of questions and answers has been decreasing over time and since the end of 2020, the number of answers has decreased below the number of questions. During this period, mobile-related post counts compared to all posts ranged from 8% to 14% per month, as shown in Fig. A2. The significant decrease in the quantity of posts observed in the latter period of 2020 can be attributed to the onset of the COVID-19 pandemic, as evidenced by Figs. A1 and A2.

## Topic modeling using LDA

At this stage, we conducted a semantic content analysis on SO mobile posts using Latent Dirichlet Allocation (LDA), a probabilistic approach for topic modeling, in order to reveal the most common issues faced by mobile developers. In text mining and natural language processing research, topic modeling provides a systematic methodology to discover the latent semantic structure of a document collection. In this respect, a number of topic modeling approaches are available such as Latent Dirichlet Allocation (LDA), Latent Semantic Indexing (LSI), Hierarchical Dirichlet Process (HDP), Non-Negative Matrix Factorization (NMF), and Dirichlet Multinomial Regression (DMR) (*Gurcan & Cagiltay, 2022*). Among the topic modeling approaches, LDA is a generative model, whose capability and efficiency is widely accepted for research based on semantic text mining, and therefore it is extensively preferred in software engineering research (*Silva, Galster & Gilson, 2021*; *Gurcan et al., 2022b*). In addition, a remarkable body of work used LDA to implement topic modeling on a number of sub-contexts of Stack Overflow (*Silva, Galster & Gilson, 2021*). LDA discovers the topics by combining words that tend to coexist commonly in text documents within the experimental *corpus* and that together form a semantic integrity (*Blei, 2012*). It uses the frequencies of words in documents and the co-occurrence of frequencies in order to create a topic model of related words. The LDA model also provides a number of well-organized methods for estimating the optimal number of topics, calculating the coherence score of discovered topics, and optimizing the topic-term distribution (*Blei, Ng & Jordan, 2003*).

Therefore, the LDA model was used in this study for the topic modeling analysis of our experimental *corpus* containing a very large number of mobile-related posts on Stack Overflow. In the following, we describe how the LDA model was fitted and implemented to our *corpus*. Initially, the preprocessing steps necessary to increase the success of the topic modeling analysis were implemented to the *corpus* (*Gurcan, 2023*). In the first step, we included only the title of the question posts in our *corpus* for topic modeling analysis by disregarding other metadata other than the title (*Rosen & Shihab, 2016*). Because, the titles are the part that best demonstrate the focal points and concepts of the issue emphasized in the posts. On the other hand, the body of the questions may contain extra information that is irrelevant to the main idea of the question (the questioner's previous experiences and

comments with the problem, previously tried methods and code snippets, other factors that triggered the problem, and so on) (*Rosen & Shihab, 2016*). This extra information creates noise in the empirical data. Consequently, as we focused on what issues developers were asking about, we excluded the body of the questions as well as the answer posts, and created a *corpus* containing only the title of the question posts for the topic modeling analysis (*Rosen & Shihab, 2016*). In the second step, tokenization, lowercase conversion, deleting numbers and punctuation, deleting stop words, and lemmatization were implemented on this *corpus* using Gensim (*Řehůřek & Sojka, 2011*), a pure Python library. Thus, data preprocessing has been completed and the empirical *corpus* has been adapted to the appropriate form essential for LDA-based topic modeling analysis (*Řehůřek & Sojka, 2011*; *Gurcan et al., 2023*).

In the topic modeling stage, we used the Gensim (*Řehůřek & Sojka, 2011*), a pure Python library developed for text preprocessing and topic modeling, to implement the LDA-based topic model to our *corpus*. Firstly, the values of the prior parameters (α, β, and K) were specified to fit and optimize the LDA model to the empirical *corpus*. The value of α parameter, which indicates the distribution of topics in documents, was used as α = 0.1, and the value of β parameter, which indicates the distribution of words in the topics, was used as β = 0.01, considering previous work on the topic modeling of short texts (*Zuo et al., 2016*; *Vayansky & Kumar, 2020*; *Gurcan & Cagiltay, 2022*). The other parameter used to obtain the ideal model was the K parameter, which indicates the number of topics. The higher the K value, the more fine-grained topics are obtained, while the lower the K value, the more coarse-grained topics are obtained. With the aim of choosing the ideal number of topics, the LDA model was implemented with various K values in the range of K ∈ {10, 11, 12, ..., 50}. Concurrently with this process, a coherence score ($C_V$) was calculated for each topic model implemented for each K value (*Řehůřek & Sojka, 2011*). As a result, a maximum coherence score ($C_V = 0.4189$) was obtained for the number of topics K = 36 (see Fig. A3), which reveals the optimal topic-word allocations for each document.

## CASE STUDY AND RESULTS

### RQ1: what issues are asked about mobile development?

As a result of the LDA-based analysis, 36 topics were discovered, in which each topic was described by 15 descriptive keywords. After examining the consistency of the topics, each topic was named taking into account the descriptive keywords of the topics. Then, we calculated the percentages of each topic in the entire *corpus*, considering the dominant topic to which each document was assigned. For example, if a topic has a 5% rate, 5% of all question posts are assigned to that topic.

The 36 topics discovered by LDA-based topic modeling, with their names, descriptive keywords, and rates are presented in Table 1. The topics in Table 1 illustrate main issues specific to mobile development, so the terms topic and issue are used interchangeably throughout this article. As seen in Table 1, the topics (issues) are listed in descending order by their percentages. Accordingly, "Android Studio", "Kotlin", and "Arrays" emerged as the top three most frequently asked topics, respectively. On the other hand, "Xamarin", "Dialog Alerts", and "Testing" were the least asked topics.

**Table 1 The 36 topics discovered by LDA.**

| Topic name | Top LDA keywords | (%) |
|---|---|---|
| Android studio | Android studio library project java gradle fail task dependency support import kotlin version module add | 4.95 |
| Kotlin | Class kotlin variable type property swift parameter function object pass extension access generic define protocol | 4.45 |
| Arrays | Value swift array specific element get number object key filter check base index contain dictionary | 4.42 |
| Error handling | Error find try get fix name give exception throw undefined module symbol solve duplicate resolve | 3.93 |
| Authentication | User app google firebase store login play facebook apple android log authentication sign get account | 3.79 |
| View controller | View scroll swiftui controller animation swift position prevent move viewcontroller present visible scrollview transition | 3.74 |
| Web view | Ionic open app webview url link web mobile cordova android page native load browser javascript | 3.57 |
| Imaging | Image load android loading imageview draw bitmap upload effect part cache rotate gallery drawable background | 3.45 |
| API requests | JSON datum request server api get response parse post fetch send retrofit swift object model | 3.34 |
| Flutter | Flutter widget camera take dart photo provider picture contact add package container get inside future | 3.34 |
| Project building | Build xcode fail error apk project version generate install framework release code target see package | 3.28 |
| List view | Item list recyclerview listview view select adapter add android inside spinner get card recycler click | 3.07 |
| Emulator | Android run application app studio emulator permission window mac install command macos process visual start | 2.98 |
| Button actions | Button screen click back action keyboard go disable press appear android tap compose open jetpack | 2.97 |
| Text settings | Text display different textview input edittext field android font label attribute character focus edit tag | 2.92 |
| Casting | String return type null convert value swift format expect empty date int cast get object | 2.81 |
| Layout settings | Layout size content height dynamic set constraint width uiview inside programmatically swift add auto uicollectionview | 2.80 |
| React-native | Reactnative component react render expo navigation child parent flatlist pass redux prop inside textinput hook | 2.80 |
| Style-theme | Change set color background default android date picker language programmatically apply style icon theme day | 2.78 |
| Event issues | IOS issue event swift detect iphone trigger simulator touch ipad fire face rotation spritekit safari | 2.77 |
| App crash | App crash problem possible android keep close difference launch setting force simple try provide stick | 2.69 |
| Connection | Device android connect phone app bluetooth connection mode enable network available mobile get ble check | 2.62 |
| File settings | File access android reference path download folder read storage local directory get pdf write object | 2.48 |
| Fragment activity | Create activity fragment android intent start pass replace viewpager context inside main unable trouble get | 2.43 |
| Navigation bar | Bar remove navigation tab bottom icon menu add hide search top space right status title | 2.31 |
| Database tasks | Update cell table sqlite row database delete tableview datum room swift exist column realm insert | 2.16 |
| Map API | API map google location android get current place memory marker point level performance delay coordinate | 2.12 |
| Notifications | Notification push send message receive app firebase android fcm local cloud background unexpected token broadcast | 2.10 |
| Firebase | Datum firebase save database share retrieve get store android child node realtime storage information read | 2.07 |
| Media streaming | Video service android play audio stream record background player slow media sound live youtube restart | 2.06 |
| Functions | Call method function objectivec execute swift callback async get outside block delegate implementation inside handler | 1.83 |
| Threading | Handle cause thread android kotlin client server wait main complete rxjava socket coroutine asynctask spring | 1.79 |
| Cloud firestore | Multiple time firestore result id get loop single query document arraylist avoid group combine cloud | 1.42 |
| Xamarin | Xamarin form page c# switch binding stack navigation drawer control xamarinform previous refresh navigator reset | 1.39 |
| Dialog alerts | Show custom android dialog ad admob listener alert unity checkbox design onclick box preview add | 1.24 |
| Testing | Code test line ui xml android system source unit allow specify break output testing write | 1.13 |

The discovered topics indicated that mobile developers faced a wide range of issues, from app development tools to debugging, database services to UI settings, casting to threading. With the aim of understanding the main knowledge domains of mobile

| Table 2 Taxonomy of the topics. | | | |
|---|---|---|---|
| Category | Topics | Topic rate | Total rate |
| Development | Arrays | 4.42 | 36.14 |
| | Error handling | 3.93 | |
| | Authentication | 3.79 | |
| | Project building | 3.28 | |
| | Emulator | 2.98 | |
| | Casting | 2.81 | |
| | Event issues | 2.77 | |
| | App crash | 2.69 | |
| | Connection | 2.62 | |
| | Notifications | 2.10 | |
| | Functions | 1.83 | |
| | Threading | 1.79 | |
| | Testing | 1.13 | |
| UI settings | View controller | 3.74 | 27.84 |
| | Web view | 3.57 | |
| | List view | 3.07 | |
| | Button actions | 2.97 | |
| | Text settings | 2.92 | |
| | Layout settings | 2.80 | |
| | Style-theme | 2.78 | |
| | Fragment activity | 2.43 | |
| | Navigation bar | 2.31 | |
| | Dialog alerts | 1.24 | |
| Tools | Android studio | 4.95 | 16.92 |
| | Kotlin | 4.45 | |
| | Flutter | 3.34 | |
| | React-native | 2.80 | |
| | Xamarin | 1.39 | |
| Data management | File settings | 2.48 | 8.13 |
| | Database tasks | 2.16 | |
| | Firebase | 2.07 | |
| | Cloud firestore | 1.42 | |
| Multimedia | Imaging | 3.45 | 5.51 |
| | Media streaming | 2.06 | |
| Mobile APIs | API requests | 3.34 | 5.46 |
| | Map API | 2.12 | |

development, we categorized the discovered topics and found that the topics fall under the following six categories: "Development", "UI Settings", "Tools", "Data Management", "Multimedia", and "Mobile APIs" (see Table 2).

## RQ2: what are the most difficult and most popular issues for mobile development?

A question post on Stack Overflow has descriptive indicators such as the view count, answer count, accepted answer count, score, favorite count, and comment count. Leveraging these indicators, we performed a set of computational analysis to identify the difficulty and popularity of each topic. Firstly, for each topic, we identified the questions in which a topic was dominant and calculated the question count related to each topic. We then calculated the average view count for each topic (dividing the total number of views by the total number of questions). In this way, we revealed the popularity of the topics. From a similar perspective, total question count, average view count, average favorite count, average voting score, average answer count, and average accepted answer count were computed and presented in Table 3. The topics in this table are sorted in descending order by their percentages.

With the aim of revealing a more understandable landscape of the difficulty and popularity of the topics, we summarized some indicators given in Table 3. Following, we depicted the first five and last five topics in Fig. 1, taking into account the average view count (for popularity). According to Fig. 1, the top five most viewed (most popular) topics are "Flutter", "Project Building" and "Error Handling", "Android Studio", and "React-Native", respectively. On the other hand, the least viewed (least popular) topic is "Firebase", followed by "App Crash" and "Dialog Alerts".

In addition, in order to reveal the difficulty level of the topics, we showed the first five and the last five topics in Fig. 2, considering the average number of accepted answers. As seen in Fig. 2, the "Connection" topic, which has the lowest rate (0.27) according to the accepted answer count, emerges as the most difficult topic. This is followed by "Media Streaming", "Notifications", "Web View", and "Emulator", respectively. According to the average answer count (see Table 3), the three most difficult topics are "Notifications", "Connection", and "Media Streaming", which are similar to the accepted answer count. Furthermore, in order to reveal other dimensions of developers' interest in topics, we presented the voting score and favorite count for each topic in Table 3.

## RQ3: how have the trends of mobile issues changed over time?

At this stage, we will try to analyze how mobile issues have changed in the last 5 years. To achieve this, we consider the distribution of the number of questions for each topic in these years. Namely, we calculated the percentage rate of the number of questions pertaining to each topic in each year. Then, we subtracted the percentages of the topics in the previous period from the percentages in the current period. Accordingly, we calculated how much the topics changed in the current year compared to the previous year. Finally, we calculated the total temporal trend of the topics at the end of these 5 years by summing the percentage changes for each topic in each period. Overall trends and annual percentages of the topics are presented in Table 4. The topics in this table are given in descending order according to their overall trend values in the last column. Among the topics, it was observed that 11 topics had an increasing trend, seven topics had a constant trend (i.e., trend values between −0.2 and 0.2), and 18 topics had a decreasing trend. As seen in Table 4, "Flutter", "React-

**Table 3 Descriptive indicators of the topics.**

| Topic name | Rate (%) | Question count | View count | Favorite count | Voting score | Answer count | Accepted answer count |
|---|---|---|---|---|---|---|---|
| Android studio | 4.95 | 51,325 | 1,565 | 0.35 | 1.80 | 1.19 | 0.35 |
| Kotlin | 4.45 | 46,101 | 1,175 | 0.30 | 1.68 | 1.30 | 0.52 |
| Arrays | 4.42 | 45,872 | 817 | 0.20 | 0.70 | 1.34 | 0.52 |
| Error handling | 3.93 | 40,745 | 1,575 | 0.26 | 1.52 | 1.23 | 0.37 |
| Authentication | 3.79 | 39,299 | 995 | 0.32 | 1.36 | 1.03 | 0.32 |
| View controller | 3.74 | 38,803 | 892 | 0.31 | 1.19 | 1.19 | 0.42 |
| Web view | 3.57 | 37,005 | 993 | 0.24 | 1.03 | 0.99 | 0.30 |
| Imaging | 3.45 | 35,748 | 963 | 0.30 | 1.07 | 1.10 | 0.38 |
| API requests | 3.34 | 34,644 | 865 | 0.20 | 0.65 | 1.15 | 0.41 |
| Flutter | 3.34 | 34,608 | 1,949 | 0.40 | 1.91 | 1.24 | 0.42 |
| Project building | 3.28 | 34,002 | 1,711 | 0.40 | 2.13 | 1.16 | 0.33 |
| List view | 3.07 | 31,870 | 819 | 0.20 | 0.69 | 1.23 | 0.42 |
| Emulator | 2.98 | 30,936 | 1,417 | 0.33 | 1.47 | 1.08 | 0.31 |
| Button actions | 2.97 | 30,783 | 948 | 0.25 | 1.03 | 1.28 | 0.40 |
| Text settings | 2.92 | 30,280 | 931 | 0.22 | 0.99 | 1.28 | 0.42 |
| Casting | 2.81 | 29,172 | 1,276 | 0.21 | 0.97 | 1.35 | 0.51 |
| Layout settings | 2.80 | 29,071 | 1,139 | 0.27 | 1.16 | 1.31 | 0.46 |
| React-native | 2.80 | 28,990 | 1,445 | 0.25 | 1.35 | 1.18 | 0.40 |
| Style-theme | 2.78 | 28,824 | 1,238 | 0.28 | 1.31 | 1.28 | 0.43 |
| Event issues | 2.77 | 28,677 | 877 | 0.33 | 1.34 | 1.03 | 0.35 |
| App crash | 2.69 | 27,894 | 744 | 0.25 | 1.00 | 1.08 | 0.35 |
| Connection | 2.62 | 27,131 | 971 | 0.29 | 1.16 | 0.92 | 0.27 |
| File settings | 2.48 | 25,665 | 1,143 | 0.25 | 1.07 | 1.01 | 0.34 |
| Fragment activity | 2.43 | 25,202 | 846 | 0.22 | 0.88 | 1.29 | 0.40 |
| Navigation bar | 2.31 | 23,939 | 1,324 | 0.34 | 1.39 | 1.32 | 0.42 |
| Database tasks | 2.16 | 22,409 | 839 | 0.25 | 0.97 | 1.19 | 0.45 |
| Map API | 2.12 | 21,973 | 1,066 | 0.32 | 1.26 | 1.02 | 0.33 |
| Notifications | 2.10 | 21,740 | 1,045 | 0.29 | 1.16 | 0.98 | 0.30 |
| Firebase | 2.07 | 21,490 | 665 | 0.19 | 0.58 | 1.09 | 0.42 |
| Media streaming | 2.06 | 21,353 | 795 | 0.30 | 1.15 | 0.86 | 0.28 |
| Functions | 1.83 | 18,948 | 988 | 0.23 | 1.13 | 1.19 | 0.45 |
| Threading | 1.79 | 18,576 | 1,281 | 0.41 | 1.76 | 1.08 | 0.41 |
| Cloud firestore | 1.42 | 14,721 | 972 | 0.25 | 1.07 | 1.15 | 0.45 |
| Xamarin | 1.39 | 14,414 | 976 | 0.24 | 1.07 | 1.10 | 0.43 |
| Dialog alerts | 1.24 | 12,810 | 784 | 0.22 | 0.86 | 1.16 | 0.38 |
| Testing | 1.13 | 11,665 | 956 | 0.27 | 1.43 | 1.05 | 0.36 |

Native", "Kotlin", "Error Handling" and "Project Building" are the top five topics with the most increasing trend, while topics "Layout Settings", "Fragment Activity", "Event Issues", "Map API", and "Database Tasks" have the most decreasing trend.

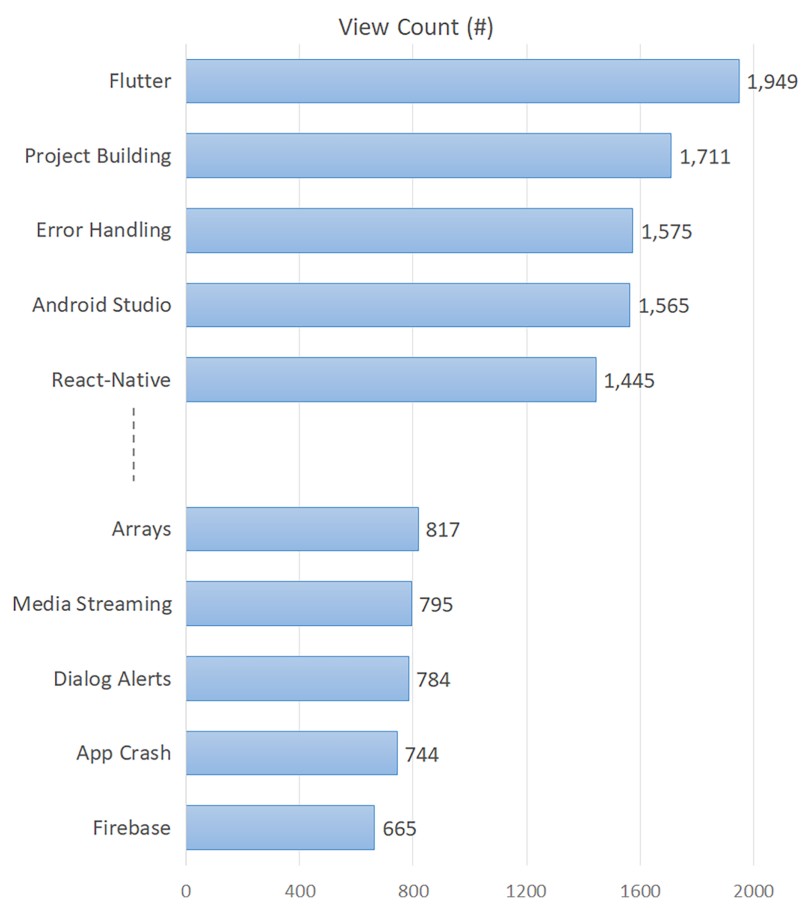

**Figure 1** Top five and last five topics by view count.

## RQ4: what are the most asked mobile technologies?

Each question post shared on Stack Overflow contains specific tags that reflect the context of the issue mentioned in that question. These tags are added to that question by the user asking the question. The tags are descriptive keywords that reveal the main themes, technologies, and tools that users associate with their questions. In order to reveal the tags related to mobile issues, we initially separated the tags of each post into singular tags and calculated the frequencies of the tags for all posts. Then, we identified the top 20 tags with the highest frequency among them. Following this, we calculated the distribution of the tags of the posts according to the topics and identified the top ten tags for each topic.

By analyzing the tags of all the posts in the *corpus*, we found that mobile developers have used 22,868 different unique tags in the last 5 years. The total number of occurrences of these unique tags was found to be 3,411,771. The average number of different unique tags used for each year was found to be 7,684. Considering the frequencies of the tags in the *corpus*, the top 20 tags with the highest frequency were identified and given in Fig. 3 in descending order by their percentages. As seen in Fig. 3, mobile technologies indicated by the tags include a wide spectrum of modules such as platforms, programming languages, development tools, and database services. Android and iOS, the two main mobile platforms, are in the first and second places, respectively. They are followed by Swift and

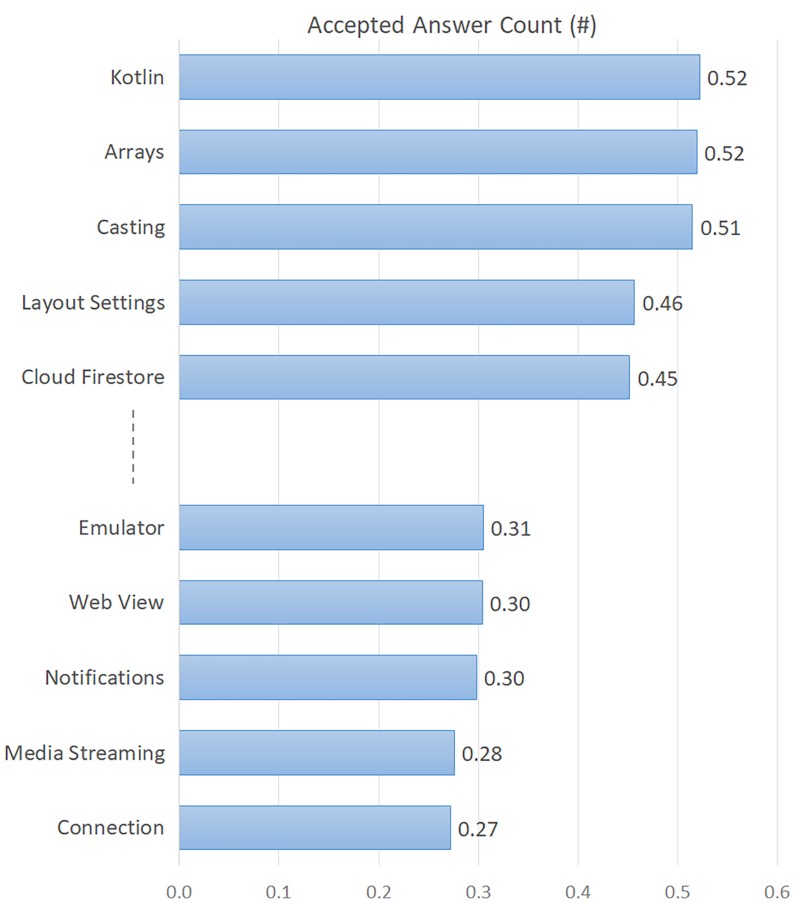

**Figure 2 Top five and last five topics by accepted answer count.**

Java programming languages. As the cross-platform mobile development tools, Flutter ranks fifth and React-Native sixth.

In the next step, with the aim of revealing the mobile technologies related to each topic, we further expanded our analysis and identified the top ten tags for each topic and presented them in Table 5. The topics are listed in descending order by their percentage in this table, where likewise the top ten tags for each topic are in descending order. In this way, we revealed a number of mobile technologies (*i.e.*, platforms, programming languages, app development tools, data services, *etc.*). As seen in Table 5, Android is featured as the first tag in 29 of 36 topics. In other words, it is seen that the Android platform is dominant in mobile problems. Although Android and iOS are seen together in many topics, iOS is in first place in only two topics ("Layout Settings" and "Event Issues").

### RQ5: how have the trends of mobile technologies changed over time?

In this trend analysis, taking into account the top 20 tags we calculated how each tag changed in that period compared to the previous period. In this way, we found the amount of increase or decrease of the tags for each year. Finally, by summing up the change amounts for each year, we found the overall trend of each tag for the last 5 years.

| Table 4  Yearly trends of the topics. | | | | | | |
|---|---|---|---|---|---|---|
| Topic name | 2017 | 2018 | 2019 | 2020 | 2021 | Trend |
| Flutter | 0.78 | 1.68 | 3.36 | 5.61 | 6.58 | 5.88 ⇑ |
| React-native | 1.64 | 2.54 | 2.95 | 3.73 | 3.65 | 2.47 ⇑ |
| Kotlin | 3.92 | 4.37 | 4.68 | 4.83 | 4.68 | 1.12 ⇑ |
| Error handling | 3.51 | 3.87 | 4.33 | 3.88 | 4.27 | 1.03 ⇑ |
| Project building | 3.2 | 3.32 | 3.37 | 3.08 | 3.51 | 0.78 ⇑ |
| Button actions | 2.98 | 2.8 | 2.93 | 2.84 | 3.34 | 0.61 ⇑ |
| Cloud firestore | 1.02 | 1.36 | 1.49 | 1.74 | 1.65 | 0.56 ⇑ |
| Authentication | 3.74 | 3.56 | 3.82 | 3.76 | 4.12 | 0.41 ⇑ |
| Android studio | 5.03 | 5.28 | 4.86 | 4.56 | 4.98 | 0.38 ⇑ |
| Threading | 1.61 | 1.86 | 1.91 | 1.82 | 1.83 | 0.36 ⇑ |
| Casting | 2.73 | 2.69 | 2.74 | 2.89 | 3.08 | 0.22 ⇑ |
| Testing | 1.11 | 1.12 | 1.17 | 1.05 | 1.19 | 0.16 ⇑ |
| Emulator | 3.09 | 2.87 | 2.79 | 2.93 | 3.21 | 0.16 ⇑ |
| File settings | 2.43 | 2.47 | 2.46 | 2.46 | 2.59 | 0.14 ⇑ |
| Navigation bar | 2.33 | 2.29 | 2.29 | 2.35 | 2.27 | 0.01 ⇑ |
| App crash | 2.73 | 2.64 | 2.71 | 2.7 | 2.67 | 0.00 ⇑ |
| Xamarin | 1.45 | 1.39 | 1.39 | 1.49 | 1.21 | −0.02 ⇓ |
| Style-theme | 2.76 | 2.72 | 2.83 | 2.85 | 2.77 | −0.03 ⇓ |
| Connection | 2.90 | 2.69 | 2.51 | 2.34 | 2.53 | −0.26 ⇓ |
| Firebase | 2.15 | 2.19 | 1.98 | 2.04 | 1.93 | −0.27 ⇓ |
| Media streaming | 2.22 | 2.11 | 1.93 | 1.91 | 2.05 | −0.34 ⇓ |
| Dialog alerts | 1.37 | 1.28 | 1.18 | 1.14 | 1.14 | −0.36 ⇓ |
| Functions | 1.90 | 1.84 | 1.84 | 1.90 | 1.62 | −0.49 ⇓ |
| Text settings | 3.11 | 3.02 | 2.98 | 2.76 | 2.63 | −0.50 ⇓ |
| Notifications | 2.33 | 2.24 | 1.99 | 1.87 | 1.92 | −0.72 ⇓ |
| Web view | 3.98 | 3.68 | 3.55 | 3.15 | 3.31 | −0.72 ⇓ |
| View controller | 3.93 | 3.64 | 3.72 | 3.96 | 3.37 | −0.79 ⇓ |
| API requests | 3.49 | 3.49 | 3.40 | 3.26 | 2.96 | −0.89 ⇓ |
| Imaging | 3.82 | 3.65 | 3.37 | 3.17 | 3.02 | −0.96 ⇓ |
| List view | 3.35 | 3.29 | 3.10 | 2.90 | 2.57 | −1.00 ⇓ |
| Arrays | 4.67 | 4.63 | 4.40 | 4.40 | 3.87 | −1.01 ⇓ |
| Database tasks | 2.50 | 2.42 | 2.16 | 1.91 | 1.62 | −1.05 ⇓ |
| Map API | 2.57 | 2.29 | 2.10 | 1.73 | 1.69 | −1.06 ⇓ |
| Event issues | 3.46 | 2.89 | 2.49 | 2.39 | 2.31 | −1.18 ⇓ |
| Fragment activity | 2.86 | 2.63 | 2.39 | 2.19 | 1.86 | −1.25 ⇓ |
| Layout settings | 3.31 | 3.2 | 2.81 | 2.41 | 2.01 | −1.41 ⇓ |

Our findings on trends in mobile technologies include annual percentages of the top 20 mobile technologies and their trend values, which are presented in Table 6. In this table, the mobile technologies are sorted by the trend values in descending order. Android and iOS, the two main mobile platforms with the highest rates, stand out as the ones with the

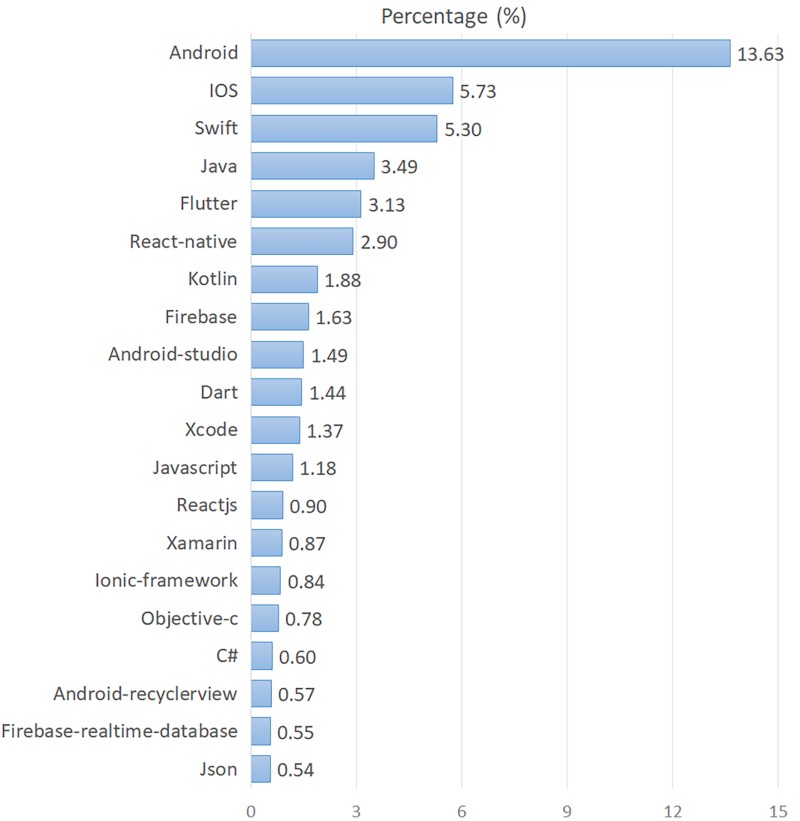

**Figure 3 Top 20 most discussed mobile technologies.**

most decreasing trend. It is clearly seen in Table 6 that trends of cross-platform development tools (*e.g.*, Flutter, Dart, Kotlin, React-native) tend to increase significantly, while native development tools (*e.g.*, Swift, Objective-C) tend to decline.

## DISCUSSION

### The wide spectrum of principal issues in domain-specific contexts

Mobile software engineering has highly dynamic and competitive working environments where paradigms, tools, technologies, skills, and experiences are constantly changing and evolving (*Rosen & Shihab, 2016*). Our analysis revealed the issues and challenges most discussed by mobile developers as 36 separate topics. The findings of our analysis clearly showed that mobile development encompasses a wide spectrum of principal issues and challenges in six domain-specific contexts including "Development", "UI settings", "Tools", "Data Management", "Multimedia", and "Mobile APIs". In order to compare the topics from our analysis with those from other studies (*Linares-Vásquez, Dit & Poshyvanyk, 2013*; *Beyer & Pinzger, 2014*; *Rosen & Shihab, 2016*; *Fontão et al., 2018*), we presented a comparative list of topics revealed by these studies, in Table 7.

Considering the results in Table 7, we found that the topics of "Layout Settings", "Database Tasks", "Media Streaming", "Error Handling", "View Controller", "Web View", "API Requests", "Casting", and "Fragment Activity" were discussed in at least three studies. In this way, these nine topics were featured as the most focused mobile issues. On

**Table 5 Related tags of the topics.**

| Topic name | Related tags |
|---|---|
| Android studio | Android android-studio java gradle kotlin android-gradle-plugin flutter react-native ios firebase |
| Kotlin | Android swift kotlin ios java flutter dart generics react-native xcode |
| Arrays | Swift android ios arrays java react-native flutter firebase kotlin javascript |
| Error handling | Android react-native ios java flutter swift android-studio firebase xcode javascript |
| Authentication | Android ios firebase swift firebase-authentication flutter react-native java google-play in-app-purchase |
| View controller | Swift ios android swiftui xcode react-native java animation uiviewcontroller uitableview |
| Web view | Android ionic-framework ios javascript cordova angular webview html react-native swift |
| Imaging | Android ios swift java flutter react-native image bitmap kotlin xcode |
| API requests | Android swift json ios java flutter react-native retrofit2 retrofit kotlin |
| Flutter | Flutter dart android flutter-layout ios firebase camera swift google-cloud-firestore flutter-dependencies |
| Project building | Android ios xcode swift react-native flutter android-studio cocoapods cordova gradle |
| List view | Android android-recyclerview java listview kotlin flutter firebase c# react-native dart |
| Emulator | Android android-studio java ios flutter react-native android-emulator macos swift xamarin |
| Button actions | Android ios swift java react-native flutter kotlin android-jetpack-compose android-studio dart |
| Text settings | Android ios swift java react-native flutter android-edittext textview android-layout xcode |
| Casting | Android swift ios flutter java dart kotlin firebase json react-native |
| Layout settings | IOS swift android uitableview android-layout uicollectionview xcode autolayout flutter java |
| React-native | React-native reactjs javascript react-navigation android expo redux react-redux ios react-native-android |
| Style-theme | Android ios swift java react-native flutter kotlin android-layout android-studio xcode |
| Event issues | IOS swift android xcode iphone react-native objective-c sprite-kit javascript flutter |
| App crash | Android ios java swift react-native flutter firebase xcode android-studio kotlin |
| Connection | Android ios bluetooth java bluetooth-lowenergy swift react-native flutter android-studio kotlin |
| File settings | Android java ios swift flutter android-studio kotlin react-native firebase file |
| Fragment activity | Android java android-fragments kotlin android-activity android-intent android-studio android-viewpager fragment |
| Navigation bar | Android ios swift flutter react-native java android-layout dart xcode kotlin |
| Database tasks | Android swift ios uitableview sqlite android-room java android-sqlite realm kotlin |
| Map API | Android google-maps ios swift java flutter react-native kotlin android-studio location |
| Notifications | Android firebase ios push-notification firebase-cloud-messaging swift notifications java react-native flutter |
| Firebase | Android firebase firebase-realtime-database java swift ios flutter google-cloud-firestore react-native dart |
| Media streaming | Android ios swift java audio flutter video react-native avfoundation android-mediaplayer |
| Functions | Android swift ios java flutter objective-c react-native kotlin dart firebase |
| Threading | Android kotlin java swift ios kotlin-coroutines rx-java rx-java2 multithreading spring-boot |
| Cloud firestore | Android google-cloud-firestore firebase flutter swift java ios dart kotlin react-native |
| Xamarin | Xamarin xamarin.forms c# android xamarin.android ios xamarin.ios react-native xaml flutter |
| Dialog alerts | Android java admob ios swift flutter android-studio kotlin unity3d android-alertdialog |
| Testing | Android java ios kotlin unit-testing swift flutter react-native android-studio android-espresso |

the other hand, topics "Arrays", "Emulator", "Button Actions", "Text Settings", "Style-Theme", "Event Issues", "File Settings", "Firebase", "Threading", and "Dialog Alerts" were covered in only one of these four studies. Unlike other studies, topics "Android Studio", "Kotlin", "Flutter", "React-Native", "Functions", "Cloud Firestore", "Xamarin", and

Table 6 Temporal trends of the top 20 mobile technologies.

| Mobile tools | 2017 | 2018 | 2019 | 2020 | 2021 | Trend |
|---|---|---|---|---|---|---|
| Flutter | 0.09 | 1.05 | 3.19 | 5.59 | 7.23 | 7.53 ⇑ |
| Dart | 0.05 | 0.57 | 1.57 | 2.51 | 3.17 | 3.36 ⇑ |
| Kotlin | 0.60 | 1.45 | 2.25 | 2.65 | 3.09 | 3.25 ⇑ |
| React-native | 1.55 | 2.56 | 3.39 | 3.65 | 3.97 | 3.01 ⇑ |
| Reactjs | 0.43 | 0.77 | 0.88 | 1.30 | 1.33 | 1.01 ⇑ |
| Android-studio | 1.04 | 1.53 | 1.25 | 1.81 | 1.98 | 0.61 ⇑ |
| Javascript | 0.96 | 1.07 | 1.19 | 1.36 | 1.41 | 0.52 ⇑ |
| Firebase | 1.37 | 1.73 | 1.56 | 1.80 | 1.76 | 0.37 ⇑ |
| Android-recyclerview | 0.51 | 0.65 | 0.65 | 0.57 | 0.49 | −0.06 ⇓ |
| Xcode | 1.43 | 1.33 | 1.33 | 1.49 | 1.23 | −0.25 ⇓ |
| Json | 0.65 | 0.57 | 0.51 | 0.48 | 0.43 | −0.25 ⇓ |
| C# | 0.71 | 0.65 | 0.65 | 0.53 | 0.42 | −0.26 ⇓ |
| Firebase-realtime-database | 0.67 | 0.67 | 0.52 | 0.44 | 0.39 | −0.29 ⇓ |
| Ionic-framework | 0.96 | 1.00 | 0.92 | 0.66 | 0.60 | −0.30 ⇓ |
| Xamarin | 1.04 | 0.99 | 0.89 | 0.78 | 0.57 | −0.40 ⇓ |
| Java | 3.67 | 3.78 | 3.79 | 3.27 | 2.83 | −0.91 ⇓ |
| Objective-c | 1.48 | 0.82 | 0.54 | 0.45 | 0.32 | −1.49 ⇓ |
| Swift | 5.53 | 5.78 | 5.67 | 5.15 | 4.16 | −1.77 ⇓ |
| iOS | 7.54 | 6.35 | 5.36 | 4.64 | 3.95 | −3.85 ⇓ |
| Android | 15.53 | 15.23 | 13.96 | 11.53 | 10.99 | −4.81 ⇓ |

"Testing" are only featured in our current study. Therefore, these eight topics can be seen as relatively new emerging issues and challenges of mobile development in recent years. These eight topics are important justifications that clearly demonstrate the widespread tendency towards cross-platform development, mobile development IDEs, and cloud-based data services.

## Insight into the use of mobile platforms, tools, and technologies

Mobile developers effectively use a wide-ranging collection of platforms, tools, and technologies covering programming languages, SDKs, IDEs, frameworks, APIs, databases, data services, and cloud-based resources in order to develop mobile apps in a more proficient way (*Jabangwe, Edison & Duc, 2018*). Our findings provide notable implications for time-dependent trends in mobile development issues, tools, and technologies (see Tables 4 and 6). Mobile development is a dynamic area where the technologies and tools used are constantly updated. Therefore, some paradigms, tools, and technologies used by the developers remain up-to-date over the years, while others become outdated in a very short time. As can be understood from our findings, while we have witnessed the dominance of Android and iOS among mobile platforms in the last 5 years, we have experienced the withdrawal of other platforms such as Windows-Phone. When evaluating our analysis and interpreting the results, it is necessary to take into account the fact that

**Table 7 Common issues in the current study and previous studies.**

| Current study | Fontão et al. (2018) | Rosen & Shihab (2016) | Beyer & Pinzger (2014) | Linares-Vásquez, Dit & Poshyvanyk (2013) |
|---|---|---|---|---|
| Android studio | | | | |
| Kotlin | | | | |
| Arrays | | ✓ | | |
| Error handling | ✓ | ✓ | | ✓ |
| Authentication | ✓ | ✓ | | |
| View controller | ✓ | ✓ | | ✓ |
| Web view | | ✓ | ✓ | ✓ |
| Imaging | | ✓ | | ✓ |
| API requests | | ✓ | ✓ | ✓ |
| Flutter | | | | |
| Project building | ✓ | | | ✓ |
| List view | ✓ | ✓ | | |
| Emulator | | | ✓ | |
| Button actions | ✓ | | | |
| Text settings | | ✓ | | |
| Casting | ✓ | ✓ | | ✓ |
| Layout settings | ✓ | ✓ | ✓ | ✓ |
| React-native | | | | |
| Style-theme | | ✓ | | |
| Event issues | ✓ | | | |
| App crash | | ✓ | | ✓ |
| Connection | | ✓ | ✓ | |
| File settings | | ✓ | | |
| Fragment activity | ✓ | ✓ | ✓ | |
| Navigation bar | | ✓ | ✓ | |
| Database tasks | ✓ | ✓ | ✓ | ✓ |
| Map API | | ✓ | | ✓ |
| Notifications | ✓ | ✓ | | |
| Firebase | ✓ | | | |
| Media streaming | ✓ | ✓ | ✓ | ✓ |
| Functions | | | | |
| Threading | | ✓ | | |
| Cloud firestore | | | | |
| Xamarin | | | | |
| Dialog alerts | | ✓ | | |
| Testing | | | | |

fewer new questions are asked about combined technologies or general framework topics, as answers to many previously asked questions are already available on Stack Overflow. Considering the top increasing and decreasing trends, our findings make it clear that certain topics and tags have reached saturation. For example, although Android and iOS

have the highest percentages (see Fig. 3), they have the most decreasing trends (see Table 6). In fact, fewer new questions can be asked on older topics. This inference is not because the topics have diminished in importance, but because many of the questions on older topics have already been answered, and repetition of questions that already exist on Stack Overflow is not allowed. Consistent with this inference, it seems that more questions are asked on topics related to newer technologies (*e.g.*, "Flutter", "React-Native", "Kotlin", and "Android Studio", see Tables 4 and 6), as they are new and most of the questions have not been asked before (*Biørn-Hansen et al., 2020*). Many of the increasingly trending topics presented in Tables 4 and 6 are fairly new technologies that became popular after 2015.

Our findings also indicated that Flutter, React-Native, Xamarin, Ionic, and Cordova are the most used cross-platform development tools for mobile apps (see Fig. 3). The findings reveal a remarkable progress from native app development to cross-platform development. The strongest proof of this insight is that the top two topics with the highest increasing trend are "Flutter" and "React-Native", respectively (see Table 4). Because, "Flutter" and "React-Native" are considered as the two most effective tools for cross-platform development. Contrasting other studies (*Linares-Vásquez, Dit & Poshyvanyk, 2013*; *Beyer & Pinzger, 2014*; *Rosen & Shihab, 2016*; *Fontão et al., 2018*), topics such as "Flutter", "React-Native", "Xamarin" first appeared in our current study (see Table 7). As a result, these empirical findings make it clear that today's mobile developers are increasingly embracing cross-platform development over time. Android Studio, Xcode, and Xamarin are the most preferred mobile IDEs (see Fig. 3). Swift, Java, Kotlin, Dart, Javascript, Objective-C, and C# are the most popular programming languages for mobile development (see Fig. 3).

Another remarkable finding of our analysis is the transition from traditional databases to cloud-based data services. Mobile developers are increasingly utilizing Firebase and its extensions, such as Google-Cloud-Firestore and Firebase-Realtime-Database (*Ozyurt et al., 2022*). As indicated by our findings, the topics of "Firebase" and "Cloud Firestore" (see Table 1) are closely related to cloud-based data services. Also, it was seen that firebase-realtime-database tag is among the top 20 most used tags (see Fig. 3). One of the important findings of our study is that mobile developers faced wide-ranging issues based on UI design, development, and its usability. In line with this background, we identified ten topics under the "UI Settings" category. The broad scope of these UI-related topics highlighted the importance of UI design, development, and optimization for mobile apps, which has also been discussed in a number of previous studies (*Punchoojit & Hongwarittorrn, 2017*; *Taba et al., 2017*). In particular, the topic of "Layout Settings" was emphasized in all of the studies compared in Table 7. These studies also indicated the necessity of "View Controller", "Web View", and "Fragment Activity" topics for UI settings. Our extensive findings, supported by other studies, potentially indicated that UI design and development is a common issue for mobile developers across various platforms (*Punchoojit & Hongwarittorrn, 2017*; *Al-Razgan et al., 2021*). In our analysis, "API Requests" and "Map API" emerged as the two main topics that revealed mobile API issues. Besides, a number of studies strongly highlighted the problems of mobile developers regarding the topic of "API Requests" (see Table 7). With regard to imaging and streaming

issues, we identified two topics including "Imaging" and "Media Streaming". In particular, the topic of "Media Streaming" was also specifically discussed in all of the studies we compared in Table 7. Among the topics discovered, "Media Streaming" has emerged as the second most difficult topic since the number of accepted answers on this topic is very low. Based on this finding, we can say that the topic of "Media Streaming" contains relatively difficult problems for mobile developers to solve compared to other problems.

## Implications for researchers and practitioners

Insights and implications from the analysis of posting data shared on Stack Overflow and similar Q&A websites can provide motivation for researchers and practitioners to create solutions for the prominent problems of mobile development. The empirical background, methodology, and findings of this study can serve as a guide for mobile development communities with diverse profiles, such as developers, researchers, practitioners, educators, and enthusiasts, to understand and contribute to the field. Revealing the wide range of development challenges faced by mobile developers, our findings provide important insights for researchers into potential research gaps that could address these issues. Each of the 36 topics discovered can be prioritized by the researchers according to the rate at which they are asked, viewed, and answered and can be considered a sub-research topic in its own context. For example, an experimental study focusing on the problems of mobile developers only in the context of UI development, taking into account the topics about UI settings (*e.g.*, "View Controller", "Web View", "List View", "Button Actions", "Text Settings," and so on), can be carried out using Stack Overflow data. Although each of the mobile development issues discovered deals with a different problem in its own right, we conclude from our findings that field researchers should prioritize, especially the most viewed and priority issues awaiting solutions. Research aimed at solving the prominent problems of mobile development can take into account the most viewed (*e.g.*, "Flutter", "Project Building", and "Error Handling") or most difficult issues (*e.g.*, "Connection", "Media Streaming", and "Notifications") raised by our findings. One potential approach to tracking the evolution and progression of mobile development trends in the future is doing periodic iterations of the present study at more frequent intervals. The methodology we have developed can also be utilized by researchers for doing experimental analysis on various textual settings.

Practitioners can contribute to the development and innovation of the field by creating useful tools and applications to solve the dominant problems of mobile development revealed by our findings. Mobile developer candidates who are new to the field can pursue a career in these areas by considering which areas have talent gaps and which topics and tools are popular. For example, "Flutter", "Android Studio", and "React-Native", which are in the top five of the most popular (most viewed) topics, offer an important perspective for practitioners on which development tools they should focus on. It can create more supportive libraries, frameworks, or guidelines for such development tools that practitioners commonly use. The "Firebase" and "Cloud Firestore" topics, which are closely related to cloud data services, highlight the need for practitioners to focus on cloud-based data services rather than traditional databases to develop data-driven mobile apps

and services. Another of our implications for practitioners that needs to be highlighted is that the top two trending topics, "Flutter" and "React-Native", point to remarkable progress from native app development to cross-platform development. Furthermore, tool developers can use our findings to fine-tune existing problematic development tools and provide more effective support and documentation. For example, the fact that "Android Studio" is the first among the discovered topics reveals the need for tool support for Android developers. In this context, it emerges as an important requirement for practitioners to prepare new helpful tools and guiding documentation for Android developers on problematic matters.

As a final word, educators can use our findings for developing curricula and training strategies that are in line with current mobile development trends. Enthusiasts and general readers within the mobile development ecosystem can refer to our findings to follow emerging developments and trends in the mobile development industry and communities. Stack Overflow and other Q&A platforms can also leverage our analysis to better categorize and tag user posts based on a more structured taxonomy. We hope that our work will guide future additive research in this area.

## CONCLUSIONS

In this study, we analyzed mobile-related posts shared on Stack Overflow using semantic text mining and LDA-based topic modeling to identify the most common issues and challenges for mobile development. In addition, we investigated the most popular mobile technologies and their temporal tendencies. The findings of our study revealed that mobile developers most frequently asked questions related to six main categories, which included "Development", "UI settings", "Tools", "Data Management", "Multimedia", and "Mobile APIs". The topics of "Flutter", "Project Building" and "Error Handling" emerged as the most popular topics. On the other hand, the most difficult topics were "Connection", "Media Streaming", and "Notifications". Our study also found that Android and iOS are the most used two platforms for mobile development. One of the key findings of our analysis was the observation of a strong transition from native app development to cross-platform development. Another notable finding was the rapid movement from traditional databases to cloud-based data services. Our analysis provides many insights into the up-to-date perspectives, issues, and needs of mobile developers. Our findings can help researchers, practitioners, and educators by revealing a wide spectrum of issues faced by mobile developers over time.

Like all studies, this one has some limitations. Initially, our research was limited to the Stack Overflow dataset. Although Stack Overflow is one of the most widely used Q&A sharing web sites among developers, it should be highlighted that focusing on a single data source may limit the scope of outcomes. Second, our analysis includes post-data provided between 2017 and 2021, and the results of our trend analysis are exclusively based on the timeline of the Stack Overflow dataset we're working with. Another limitation of our study is that, as in other clustering techniques, the process of identifying the topic labels is subject to the authors' perspective and interpretation of the results. Fourth, the topic estimation parameters for the LDA topic modeling approach used in this study may vary depending

on the data type and context used. Finally, because our study incorporates inductive and probabilistic exploratory procedures, future confirmatory research is required to test and enhance our findings.

Future work can extend this study in many avenues. Researchers can leverage our methodology to analyze trending topics and movements specific to the different research contexts they are interested in. The methodology employed in this study can be extended to encompass further developer Q&A web sites, such as Kaggle, Reddit, GitHub, and Quora. Our methodology can be applied to other data resources such as web portals, social networks, developer blogs and forums, and compare our findings for compatibility with those in these environments. Different data processing methods, preprocessing stages, and semantic text mining approaches can be joined to develop new hybrid models. The present methodology can be enhanced with new supportive approaches for topic discovery in different domains. Studies planned for the future will extend our methodology using different topic modeling approaches such as Hierarchical Latent Dirichlet Allocation (HLDA), Hierarchical Dirichlet Process (HDP), Non-Negative Matrix Factorization (NMF), and Dirichlet Multinomial Regression (DMR).

### Funding
The authors received no funding for this work.

### Competing Interests
The authors declare that they have no competing interests.

### Author Contributions
- Fatih Gurcan conceived and designed the experiments, performed the experiments, analyzed the data, performed the computation work, prepared figures and/or tables, authored or reviewed drafts of the article, and approved the final draft.

### Data Availability
The Stack Overflow Post Repository is available at: https://archive.org/download/stackexchange.

A processed sample of the experimental data is available in the Supplemental File.

### Supplemental Information
Supplemental information for this article can be found online at http://dx.doi.org/10.7717/peerj-cs.1658#supplemental-information.

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
