# Peer review of "Identification of mobile development issues using semantic topic modeling of Stack Overflow posts"

_PeerJ Computer Science, doi:10.7717/peerj-cs.1658_

## Round 0.1 · original submission · Major Revisions

Both reviewers think that the paper needs major changes to be considered for publication. Please address all the comments.

·

Basic reporting

In this manuscript, the authors analyzed mobile-related posts based on topic modeling (LDA) on Stack Overflow to extract the most common issues and challenges for mobile development. Although, the structure of the paper is well, however I would like to suggest some comments:
- Regarding the research model, there are several topic modeling methods. What is the intuitive idea behind LDA?
- It seems that a discussion about the role of Inference methods (such as Gibbs Sampling ) for LDA is missing, it needs to be highlighted.
- The limitation work is not clear, and lack of discussion.
- Figures 1-3 are not clear!

Experimental design

Experimental design, analysis of results, discussion all are good. The obtained results for topic modeling is well described. Although, Tables 1-9 are easy to read and understand. But the font size for Figures 1-3 are not clear!

Validity of the findings

Relevant data are reported for the experiments and statistical analyses employed are appropriate.

Reviewer 2 ·

Basic reporting

The study reported in the paper is about an analysis of stack overflow posts about mobile development issues, and the application of sematic topic modeling on the mined data.

The study performed is definitely interesting since there is a significant amount of information that can be mined from platforms and Q&A websites suck as stack overflow where questions can be paired with code snippets, solutions and comments.

However, the paper is its current state cannot be accepted for publication due to significant presentation issues and the lack of a more nuanced discussion. I provide detailed comments below.




Introduction:

- quotes are needed for the sentences in the first 4 rows.
- The citations about the problems for mobile app development are too old (2013, 2014) given the evolution of mobile development in the last ten years. Please update them with more recent quotes.
- The term "android" should always be capitalized


Background:

- the authors point out that Rosen & Shihab in their 2016 study performed a very similar type of analysis to the one described in the current paper. At this point of the paper, it is unclear why performing the same analysis anew would be worthy. It is definitely true that the mobile development landscape has changed in seven years, but the author does not point this out - instead, only the diffusion of mobile applications and mobile development is mentioned. I think that to give more value to the present paper, a more thorough justification of the need for a new analysis of issues on Stack Overflow would be needed - something more convincing than just "the previous one is dating back to 2016".

- are there any alternatives to the Latent Dirichlet Allocation? The author should motivate why the method was chosen, and not only how it works.



Case Study and Results

- Under RQ1 (line 353) it is not necessary to repeat again what LDA is

- For RQ1 and for all other research questions, I would expect the tables with the result to be within the main text of the paper and not at the end of the paper as an appendix. Having the results in an appendix forces the reader to go back and forth in the paper and it causes a lot of fatigue in reading the full paper.

- For all research questions: the structure of the paper should be made better by separating the approaches for each research question and the findings. I would expect a section (called maybe "analysis method" or "approach" by using the authors' nomenclature) followed by a section with the findings only.

- For all research questions: "Finding" should be "Findings"

- RQ2 (line 406): "for each topic, we identified the questions in which a topic was dominant and calculated the question count related to each topic" - I had to read this sentence a couple of times, maybe it should be simplified to something like: "for each topic we counted the questions where it was dominant"?

- The author should emphasize that the decrease of the "Android" and "iOS" topics is actually just apparent in that the other topics are actually "within" them. For instance we have an increasing trend for Android-studio that to some extent compensates the decreasing trend of Android.

- I am puzzled by the enormous drop in figure 2. Is that really possible that we had 0 questions related to Android at the beginning of 2021?

- The visualizations for the graph should be better rendered (consistent size of fonts, units of measures in the axis). As for the tables, the figures should be integrated in the text.

- Figure 3 is pretty obscure. All figures should be self-contained so I expect a proper motivation in the figure caption and axis labeling.


Discussion

The discussion is the most shallow section of the paper and I would expect this section to be significantly deepened for the paper to be published. Currently, the discussion just re-lists the results that are provided in the tables, with no elaboration on the possible indications for practitioners and reasoning about the results that have been found.

I would expect in this section to have insights for developers, hints about the possible evolution of the market and the technologies, and actionable guidelines for researchers and developers in the Android domain. At the current moment as a reader I just find a list of topics and some information about the top techniques used (e.g. Flutter and React-Native for cross-platform development).

As a side note, it would be interesting to see how the different topics correlate with each other, but this is more for future work than for the presentation of the current findings.


Threats to validity:

- the author should mention other q&a websites alternative to stack overflow

Experimental design

No specfic issues with the experimental design

Validity of the findings

The authors should better frame the impact of the study and derive more actionable guidelines for practitioners to make them usable in practice.

---

## Round 0.2 · accepted · Accept

The paper is now ready for publication.

·

Basic reporting

The authors improved the paper based on the suggested comments. I believe that the current version can be ready for publication.

Experimental design

Satisfied

Validity of the findings

Satisfied

Reviewer 2 ·

Basic reporting

I am happy with all the authors' responses and all my comments have been addressed.

I think that the manuscript can be considered for acceptance.

Experimental design

-

Validity of the findings

-